

**OBS noise reduction from horizontal and vertical**
**components using harmonic-percussive separation**
**algorithms**
Zahra Zali[1,2], Theresa Rein[1], Frank Krüger[1], Matthias Ohrnberger[1], Frank Scherbaum[1]
[1]University of Potsdam, Institute of Geosciences, Karl-Liebknecht-Str. 24-25, 14476 Potsdam, Germany
[2]GFZ German Research Centre for Geosciences, Potsdam, Germany
*Correspondence to*: Zahra Zali (zali@uni-potsdam.de)
Zahra Zali: Campus Golm, Building 29, Room 2.48, Karl-Liebknecht-Str. 24-25, 14476 Potsdam, Germany
(zali@uni-potsdam.de)
Theresa Rein: Campus Golm, Building 27, Room 0.43, Karl-Liebknecht-Str. 24-25, 14476 Potsdam,
Germany (theresa.rein@uni-potsdam.de)
Frank Krüger: Campus Golm, Building 27, Room 1.36, Karl-Liebknecht-Str. 24-25, 14476 Potsdam,
Germany (Frank.Krueger@geo.uni-potsdam.de)
Matthias Ohrnberger: Campus Golm, Building 27, Room 1.37, Karl-Liebknecht-Str. 24-25, 14476
Potsdam, Germany (Matthias.Ohrnberger@geo.uni-potsdam.de)
Frank Scherbaum: Campus Golm, Building 29, Room 1.52, Karl-Liebknecht-Str. 24-25, 14476 Potsdam,
Germany (Frank.Scherbaum@geo.uni-potsdam.de)



**Abstract**
Records from ocean bottom seismometers (OBS) are highly contaminated by noise, which is much higher
compared to data from most land stations, especially on the horizontal components. The high energy of the
oceanic noise at frequencies below 1 Hz complicates the analysis of the teleseismic earthquake signals
recorded by OBSs.
Previous studies suggested different approaches to remove low frequency noises from the data, but mainly
focused on the vertical component. The records of horizontal components, crucial for the application of
many methods in passive seismological analysis of body and surface waves could not be much improved in
the teleseismic frequency band. Here we introduce a noise reduction method, which is derived from the
harmonic-percussive separation algorithms used in Zali et al., (2021) in order to separate long-lasting
narrowband signals from broadband transients in the OBS signal. This leads to significant noise reduction
of OBS records on both the vertical and horizontal components and increases the earthquake signal to noise
ratio without distortion of the broadband earthquake waveforms. This is proved through synthetic tests by
measuring SNR and cross-correlation coefficient where both showed significant improvement for different
realistic noise realizations. The application of denoised signals in surface wave analysis and receiver
function is discussed through synthetic and real tests.

**1 Introduction**
Data from ocean bottom recordings are commonly difficult to analyze, due to the high noise level being
typically much higher compared to land stations. At frequencies below 1 Hz, the effect of the ocean noise is
often dominating the data and hinders the seismological analysis (e.g. Webb et al., 1991; Crawford, 1994).
Signals of interest, i.e. transient signals, especially from teleseismic events can be masked by the oceanic
noise. Here, the horizontal components are most strongly contaminated by low frequency noise. To
illustrate the noise on OBS data, we exemplary show the records of the station D10 of the DOCTAR array
(see Fig. 1 and Fig. S1). Various studies tried to identify and characterize the different sources of noise
recorded at the ocean bottom (e.g. Webb, 1998; Crawford & Webb, 2000; Corela, 2014; Stähler et al.,
2018; Essing et al., 2021; An et al., 2021). We focus on noise sources that especially affect teleseismic
horizontal recordings in the frequency band of 0.02–2 Hz. Generally, the dominant natural noise signals in
the oceanic environment are secondary oceanic microseisms (Rayleigh/Scholte waves at the ocean bottom)
caused by interaction of wind generated water waves, infragravity waves (compliance noise) as well as tilt
noise; the latter is originating from the turbulent interaction between currents and the instrument (e.g.
Crawford et al., 1998; Corela, 2014). Primary oceanic microseism is originating from the interaction of
water waves incident at steep coastlines and/or rough seafloor (Hasselmann, 1963; Webb, 1998; Bell et al.,
2015) and its spectrum peak is around 0.07 Hz (Friedrich et al. 1998) in the Northern Atlantic. The
secondary microseism, in contrast shows signal frequencies above 0.1–0.25 Hz, with a maximum spectral
peak around 0.14 Hz (Friedrich et al., 1998, Fig. 1) and is in general much stronger in amplitude than the
primary microseism. The secondary microseism is caused by wind or swell waves propagating in opposite



directions. The periods of such generated Rayleigh/Scholte-waves are half the period of the water waves
generating them (e.g. Longuet-Higgins, 1950; Bell et al., 2015) and strongly affect the seismological
analysis. Whereas the primary and secondary microseisms affect both the vertical and horizontal
seismometer components, the compliance noise is solely observed on the vertical component and the
hydrophone. Compliance noise, dominant in the frequency band of 0.01–0.04 Hz, is only significant if the
wavelength exceeds the water depth (Crawford et al., 1998; Crawford & Webb, 2000; Bell et al., 2015).
Below frequencies of 0.01 Hz and 0.1 Hz, the vertical and especially the horizontal components, are highly
contaminated by tilt noise generated by ocean bottom currents (Webb, 1998; Crawford& Webb, 2000;
Stähler et al., 2018, Fig. 1). The tilt noise level increases with signal period (see Fig. 1).  The ocean bottom
currents in many regions of the oceans are mostly driven by tidal force and often create a signal with
strongest amplitudes below 1 Hz appearing every 6–12 hours (e.g. Brink, 1995; Crawford & Webb, 2000;
Ramakrushana Reddy et al., 2020; Essing et al., 2021). The ocean bottom currents passing the instrument
create local eddy currents, deform the seafloor beneath the sensor and tilt the whole instrument frame, to
which the seismometer is fixed (e.g. Duennebier & Sutton, 1995; Webb, 1998; Romanowicz et al., 1998;
Crawford & Webb, 2000; Corela, 2014; Stähler et al., 2018).  If the seismometer mass is not perfectly
leveled, the high tilt noise on the horizontal components is partially projected onto the vertical component
(e.g. Crawford, 1994; Corela, 2014; Bell et al., 2015). Since the noise sources often act at frequencies of
teleseismic earthquakes, it is crucial to improve the Signal-to-Noise Ratio (SNR) on OBS recordings for the
analysis of the earth's crustal and mantle structure. Various studies discussed the improvement of OBS
recordings through different approaches, either by suggesting a better OBS instrument design (Stähler et
al., 2018, Corela, 2014, Essing et al., 2021), or by removing significant amounts of the noise from the
contaminated data by signal processing (Crawford & Webb, 2000, Bell et al., 2015, Janiszewski et al.,
2019). Our study follows the latter approach.
Crawford and Webb (2000) developed a method to remove noise from the vertical OBS component.
Calculating the linear transfer function between the horizontal and the vertical component allows to
estimate the tilt noise which can be subtracted from the vertical component. Hydrophone data measured in
parallel to the seismometer recordings allow to reduce the influence of infragravity waves on the vertical
seismometer component recordings. For better results Bell et al. (2015) propose to first rotate the horizontal
components in direction of the highest coherence between the horizontal and vertical component before
calculating the linear transfer functions. The mentioned methods solely improve the SNR on the vertical
component whereas the noise contamination on horizontal components is often larger. Other recent studies
attempted to reduce noise also on the horizontal components (Mousavi and Langston, 2017; Zhu et al.,
2019; An et al., 2021; Negi et al., 2021). An et al. (2021) tried to reduce the noise on the horizontal
components by applying the reversed procedure of Bell et al. (2015). Rotation of one horizontal component
into the direction of the principle noise indeed results in an improvement of the orthogonal horizontal
component, but the other horizontal component became noisier (An et al., 2021). Results of a recent study
applying a polarization filter to reduce the noise on all components show strong changes of the broadband





waveforms (Negi et al., 2021). The automatic noise-attenuation method developed by Mousavi and
Langston (2017) is a time-frequency denoising algorithm using the wavelet transform and
synchrosqueezing. It can be either used to keep the signal and remove the noise or vice versa. The
decomposition method DeepDenoiser from Zhu et al. (2019) is based on a deep neural network.
DeepDenoise decomposes the waveform into signal and noise in the time-frequency domain. The latter
methods, both improve the SNR, but mainly focus on local and regional earthquake detection and result in
changes in the waveform shape if the noise amplitude directly ahead of the signal is significant in
comparison to the signal amplitude in a specific frequency. However, the analysis of undistorted broadband
waveforms on the horizontal components is crucial for many passive seismological structure analysis
methods, e.g. receiver functions or surface wave dispersion and polarization analysis.
Here we introduce a method, inspired from the music information retrieval (MIR), which is adapted to
seismological data and is used for noise reduction on both, the vertical and the horizontal components.
Seismic waveform and acoustic signals generated by musical instruments are similar in some aspects
(Schlindwein et al., 1995; Johnson and Watson, 2019). The extensive research in the field of music
information retrieval has resulted in advances (e.g., Müller, 2015) that may be useful in seismic signal
processing as well. Exploiting the idea of harmonic-percussive separation (HPS) in MIR, Zali et al. (2021)
developed an algorithm to separate harmonic volcanic tremor from earthquakes in seismic waveforms. In
this study we use this algorithm after some modifications in order to separate 'harmonic' (long-lasting
narrowband signals) and 'percussive' (broadband transients) components of an OBS data set aiming at
noise reduction and retrieval of clearer broadband earthquake waveforms. Throughout this study we will
make use of the term noise for any signal other than earthquake signal in the data set. In the context of OBS
noise reduction using HPS algorithms, percussive components correspond to earthquake signals and
harmonic components correspond to noise signals. The long-duration OBS noise signals that last a few
hours to days (depending on the noise type) with a restricted frequency range contrasts with transient
seismic signals such as earthquakes with a wider range of frequencies.
The algorithm introduced in Zali et al., (2021) is a combination of two HPS approaches that leads to the
desired signal separation. Here we also use the two approaches in order to separate different type of noise
signals from the earthquake signals. In the first step we adopt HPS using a similarity matrix (Rafii and
Pardo, 2012; Rafii et al., 2014) to separate monochromatic and harmonic noises. In the second step we
adopt HPS using median filtering (FitzGerald, 2012) in order to separate the remaining narrow-band
signals. With this two-step approach we can separate and remove much of the OBS noise contamination
from the earthquake signals.

**2 Data**
In this study we discuss the noise recorded by a LOBSTER (Longterm OBS for Tsunami and Earthquake
Research) OBS instrument from the DEPAS pool, which is equipped with a Güralp CMG-40T
seismometer, a MCS (Marine compact seismic) recorder and loose cables (for technical specification see



the DEPAS Pool web page and Alfred-Wegener-Institute, Helmholtz-Zentrum für Polar- und
Meeresforschung et al., 2017). We show data recorded during the DOCTAR deployment, using DEPAS-
LOBSTERs, located around the Gloria Fault in the Northern Atlantic. Twelve DEPAS-LOBSTERs form
the array. They were deployed between 2011-2012 and recorded the data with a sampling frequency of 100
Hz (Hannemann et al., 2016; Hannemann et al., 2017).
Until 2019 the DEPAS LOBSTER OBS was built with an OBS-specific version of the Güralp CMG-
40T/MCS recorder, where the seismometer had a corner frequency of 60 s and has been modified to last
long on the seafloor (Stähler et al., 2018). However, the development of less power consumption lead to a
higher noise level of the instrument itself (Stähler et al., 2018). At low frequencies (<0.1 Hz) the self-noise
of the sensor is highly affecting the records, especially on the vertical component. However, the design of
the DEPAS-LOBSTERs has been improved for deployments after 2019 (e.g. Essing et al., 2021).
We observed a continuous harmonic signal at a frequency of 0.04 Hz, partially with one or two overtones
on a subset of the array (see Fig. 1). This signal was observed on 30% of the stations from the DOCTAR
project (e.g., Hannemann et al, 2016, Hannemann et al., 2017) and on 43% of the stations from the
KNIPAS project (Schlindwein et al., 2018), both using the mentioned DEPAS-LOBSTER design. We
cannot identify the source of this signal yet, but based on its continuity, we assume an electronic source
from the instrument itself.
The hydrophone and especially the horizontal components are highly affected by the strumming of the
head-buoy, which is attached to the DEPAS-LOBSTER frame causing a 'current induced harmonic tremor
signal' (Stähler et al., 2018; Essing et al., 2021, Fig. 1). These 'tremor events' last over up to 4 hours and
appear every 6–12 hours. These presumably tidal-driven tremor events are harmonic signals with a
fundamental period of 0.4–1 s and various overtones (1–10 Hz) (Stähler et al., 2018; Essing et al., 2021,
Fig. 1). Regarding the frequency band, 'tremor events' mainly affect the analysis of teleseismic body
waves, especially on the horizontal component (Fig. 1).

**3 Noise reduction methodology**
**3.1 Harmonic-percussive separation (HPS)**
Harmonic-percussive separation refers to the problem of decomposing a signal into its harmonic and
percussive components. This topic has received much attention in recent years (Rafii et al., 2018) and has
numerous applications in the field of MIR and musical signal processing.
Within a general context harmonic signals show an overtone structure in the spectral domain. We call
overtones one or more clear narrow-banded frequency peaks being integer multiples of the fundamental



frequency (the first frequency peak in the spectrum). Harmonic signals have a relatively stable behavior
over time and can be identified in a Short Time Fourier Transform (STFT) spectrogram by horizontal
structures referred to constant frequencies along the time axis.
In contrast percussive signals form vertical structures in a STFT spectrogram that contain energy in a wide
range of frequencies. Therefore it is a straightforward strategy in most HPS algorithms to try to separate the
horizontal structure from the vertical structure in the spectrogram corresponding to harmonic and
percussive components, respectively. The horizontal lines in the spectrogram could correspond to either
harmonic signals or monochromatic signals.
OBS noise forms narrowband horizontal structures in the STFT spectrogram while earthquake signals have
vertical exhibition in the STFT spectrogram.
**3.2 HPS using median filtering (MED)**
In the context of HPS one of the simplest and fastest approaches is median filtering (FitzGerald, 2010). For
simplification we name this algorithm as MED in this study. Median filters are usually used to remove
noise from an image or a signal. Using median filter a sample will be replaced by the median of
neighboring samples within a window of a specific length (The specific length is the kernel size of the
median filter). The entire signal is processed using a sliding window analysis. Within the HPS, two median
filters are applied to the amplitude of the STFT spectrogram of a signal. A median filter is performed along
the time axis of the spectrogram to suppress percussive events and enhance harmonic components. Another
median filter is applied along the frequency axis in order to enhance percussive events and suppress
harmonic components. The two resulting spectrograms are then subsequently used to create two masks,
which are applied to the original signal spectrogram separately to generate two spectrograms of harmonic
and percussive components. For creating the harmonic and percussive signals in time domain the phase of
the original signal is added to each spectrogram and the time domain signals are reconstructed using the
inverse STFT.
**3.3 HPS using the similarity matrix (SIM)**
Another powerful approach in HPS proposed by Rafii & Pardo (2012) is based on calculating a similarity
matrix. We name this algorithm as SIM here. This approach is a repetition-based separation, which
identifies the repeating elements by looking for similarities by means of a similarity matrix. Within the SIM
algorithm, first similar time frames in the spectrogram are identified through a similarity matrix. Then a
median filter is applied only to the frames identified as similar to constitute the repeating spectrogram
model that corresponds to harmonic components. The non-repeating spectrogram that corresponds to the
percussive component of the data is obtained by subtracting the repeating spectrogram from the original



spectrogram. For creating the repeating and nonrepeating signals in time domain the phase of the original
signal is added to each spectrogram and the time domain signals are reconstructed using the inverse STFT.
Details of this approach are discussed in the following section.

**3.4 HPS noise reduction algorithm for OBS data**

The motivation of using HPS for noise reduction of OBS data is originated from different characteristic of
earthquake and OBS noise signals as described in Sect. 2. Earthquakes are broadband transient signals,
while most of OBS noises are dominantly narrow-band signals and can be considered to have a
monochromatic or harmonic appearance in the spectrogram. We combine two modified HPS algorithms to
separate those signals in a two-step procedure. We divide the frequency content of the signal into two
ranges; one is between 0.1 to 1 Hz and the other is everywhere out of this frequency range. In the first step
we use the SIM algorithm and separate only harmonic or monochromatic signals from the original records
everywhere out of the mentioned frequency range. In the second step we target the specified frequency
range containing harmonic (or monochromatic) as well as narrow-band signals with gliding frequencies
named as current induced harmonic tremor signal in the Sect. 2 previously. The overall schematic diagram
of our HPS noise reduction algorithm along with an example is shown in Fig. 2.
The SIM algorithm is explained in the following: From the original OBS record SO (SO represents the
original restituted OBS signal) we derive the STFT named $X$ being a complex-valued spectrogram.
The complex-valued spectrogram **X** is separated into its amplitude and phase components using Eq. 1.

$$X = V * exp(1j * \varphi), \tag{1}$$


where $\varphi$ is the phase of **X**, $V = |\mathbf{X}|$ is the amplitude of $X$ and $j$ is the imaginary unit.
All of the spectrogram modifications will be done on the amplitude spectrogram **V**. The cosine similarity
(the similarity between two vectors of an inner product space) between the STFT time frames is calculated
through the multiplication of the transposed **V** by **V** with normalization of the **V**. This is shown in Eq. 2.

$$S(k_a, k_b) = \frac{\sum_{i=1}^{n} V(i, k_a) V(i, k_b)}{\sqrt{\sum_{i=1}^{n} V(i, k_a)^2} \sqrt{\sum_{i=1}^{n} V(i, k_b)^2}}, \tag{2}$$


where **S** is the similarity matrix. Each point $(k_a, k_b)$ in **S** is the cosine similarity between time frame $k_a$ and
$k_b$ of **V**, $\forall k_{a, b} \in [1, m]$, where $m$ is the number of time frames and $n$ is the number of frequency channels
for each time frame. Once the similarity matrix is calculated we use it to determine the most similar time
frames to each single time frame. For time frame $k_a$ we compare all the values in $\mathbf{S}(k_a, k_i)$ for $i \in [1, m]$.
2% of the all time frames, which have the highest S values, are identified as similar frames for time frame



$k_a$.
Finally, all similar time frames to any frame $k$ in **V** are stored in a temporary array **K**. Those similar time
frames are used to create a repeating spectrogram model **W**. The corresponding frame in **W** is obtained by
taking the median of **K** for each frequency at each time frame $k$. Those time-frequency bins, which are
similar with little deviations between repeating frames, are captured by the median and constitute the
repeating spectrogram model. This spectrogram contains only similar and repeating patterns. The time-
frequency bins with large deviations between repeating frames would constitute nonrepeating transient
patterns and would be suppressed by the median filtering.

The nonnegative spectrogram **V** is the sum of two nonnegative spectrograms of repeating and nonrepeating
patterns, hence, **W** (the repeating spectrogram model) should always have smaller values or at most be
equal compared to **V**. To ensure this a repeating spectrogram model $\widetilde{\mathbf{W}}$ is defined by taking the minimum
between **W** and **V**. The nonrepeating spectrogram model is derived by subtracting $\widetilde{\mathbf{W}}$ from $V$.

We use these two (the repeating and the nonrepeating) spectrogram models to create two time-frequency
masks for repeating and nonrepeating patterns. Instead of the binary mask, which is used in Rafii & Pardo
(2012), we use soft masks via Wiener filtering (Vaseghi, 1996). The calculation of the soft masks is shown
in the following equations:

$$M1 = \frac{\widetilde{W}^2}{\widetilde{W}^2 + \left(V - \widetilde{W}\right)^2}, \tag{3}$$

$$M2 = \frac{\left(V - \widetilde{W}\right)^2}{\left(V - \widetilde{W}\right)^2 + \widetilde{W}^2}, \tag{4}$$



in which **M1** and **M2** are repeating and nonrepeating masks respectively. We multiply the masks with the
input amplitude spectrogram **V** to separate the repeating and nonrepeating components. The element-wise
multiplication of the masks by the input amplitude spectrogram **V** is shown in the following equations:

$$R = M1 \otimes V, \tag{5}$$

$$NR = M2 \otimes V, \tag{6}$$


in which **R** and **NR** denote repeating and nonrepeating amplitude spectrograms respectively.



The resulting **R** and **NR** spectrograms are shown in Fig. 2a for a specific/typical example of an OBS
recording. As can be observed in the **R** spectrogram, in particular the low frequency harmonic or
monochromatic signals below 0.1 Hz are well captured. We applied the SIM algorithm only to the
frequency band below 0.1 Hz and above 1 Hz.  In the frequency band from 0.1 Hz to 1 Hz the signals
remains unchanged by this procedure. This is the first constraint we consider for the SIM algorithm. The
reason is related to the frequency content of the noise and earthquake signals and how the SIM algorithm
separates them. In the field of noise reduction using signal processing techniques, a very important point is
to not modify the signals of interest for analysis. P and S waveforms in the teleseismic earthquake signals
have often frequency content in the range of 0.1 Hz to 1 Hz with a dominant frequency around 0.3 Hz.
Oceanic microseism noise, which is usually present in the OBS data, has a dominant frequency around 0.1
Hz to 0.3 Hz. As P and S phases have similar dominant frequency as the microseism noise wavefield,
superposition of both wavefields could happen in this frequency range. They could interfere constructively
or destructively so the resulting amplitude could be higher or lower compared to the original P or S phase
amplitudes. Considering these interferences, using the SIM algorithm, may result in creating fake higher
amplitude for these phases or losing part of their amplitude in the noise reduced signal. But this could be
problematic only when the amplitude of the noise is changing over the time. For a noise signal with almost
constant amplitude, the SIM algorithm can extract the true amplitude of the noise even in the interference
moments. However, the microseism noise has slightly varying amplitude over time.
Before moving to the second step we introduce a second constraint parameter, which we use in the SIM
algorithm. Surface waves of teleseismic events show usually a dispersed narrow-band signal and
correspond to (on a daily scale) short duration mainly horizontal patterns in the spectrogram. Given the way
the HPS is separating harmonic from transient signals, the surface wavetrain may be erroneously
recognized as harmonic component and thus be separated as noise signal. In order to prevent this and
preserve the whole frequency content of the earthquake, we define a so-called waiting factor for the
similarity calculation introducing a minimum time distance between two consecutive similar frames. For
the problem of retaining teleseismic surface waves we found that a waiting time of at least two hours
prevents the algorithm to prune surface waves from the transient signal part. The rationale is that the
duration of a teleseismic event is usually less than two hours whereas the noise components have longer
duration. Using this waiting factor prevents separating any harmonic component of the earthquake signal as
noise component. As a side effect this constraint causes that short duration harmonic/monochromatic noise
signals won't be well captured, too. However, these types of signals are not common in OBS data (see the
Sect. 2).
In the second step of our algorithm, to target noise signals in the frequency range of 0.1 Hz to 1 Hz, we use
MED as it is described in the Sect. 3.2. We apply this second part of the noise removal procedure only to a
restricted frequency band of 0.1 Hz to 1 Hz.  We don't apply MED for the frequency range below 0.1 Hz to





avoid an interference with the surface wave signals of teleseismic events that shall be retained. The dominant noise in the mentioned frequency range is the current induced harmonic tremor signal (see the Sect. 2).

First we create $\mathbf{X'}$ spectrogram which is equal to $\mathbf{X}$ in the mentioned frequency range and is equal to zero out of this band. Then a horizontal median filter is applied to $\mathbf{X'}$ in order to separate harmonic components. Near horizontal patterns will be captured by the median filter and will be separated in the harmonic spectrogram $\mathbf{H}$.

Now we have two separated spectrograms for noise signals: $\mathbf{R}$, which is derived from the first step, and $\mathbf{H}$, which is derived from the second step. Summing these two spectrograms will build the noise spectrogram $\mathbf{N}$. Subtracting $\mathbf{N}$ from the input amplitude spectrogram $\mathbf{V}$ will construct the transient spectrogram $\mathbf{T}$.

As can be seen in Fig. 2a in step 2, the dominant energy of the narrow-band signals with gliding frequencies in the range of 0.1 Hz to 1 Hz (the current induced harmonic tremor noise as introduced in the Sect. 2) is captured in the noise spectrogram $\mathbf{N}$, but part of that is still remained in the transient spectrogram $\mathbf{T}$. The signals with changing frequency that don't form complete horizontal lines in the spectrogram are difficult to be captured by our HPS algorithm so part of their energy remains in the final spectrogram.

### 3.5 Reconstruct the denoised signal

In order to reconstruct the noise-removed signal in time domain we must add phase information to the spectrogram. We had separated the complex-valued spectrogram $\mathbf{X}$ into its amplitude $\mathbf{V}$ and its phase component using Eq. 1 and all the further modifications have been applied to the amplitude spectrogram $\mathbf{V}$. The phase of input signal SO is mostly affected by the phase of noise signals as they have the dominant energy in the signal. Therefore we use phase information of SO in order to reconstruct the noise signal. We add this phase to the noise spectrogram $\mathbf{N}$ using the following equation:

$$N' = N * exp(1j * \varphi), \qquad (7)$$

where $\mathbf{N'}$ is the complex-valued noise spectrogram. We reconstruct the noise signal $NS$ from the complex spectrogram $\mathbf{N'}$, using the inverse STFT. Finally the OBS denoised signal HPS (HPS here represents the SO signal after the HPS processing) is obtained by subtracting the noise signal from the input OBS signal SO using the following equation:

$$HPS = SO - NS, \qquad (8)$$

### 3.6 Parameters selection



Many typical noise signals observed at OBSs are harmonic, monochromatic or narrow-band signals with
gliding frequencies (see the Sect. 2). In order to extract the expected narrowband noise signals from the
STFT we require a high frequency resolution in the spectral domain therefore making it necessary to use
sufficiently long time windows for the spectral analysis. Here we use an FFT window length of 163.84
seconds with an overlap of 75%, corresponding to an FFT size of 16384 at a sampling frequency of 100 Hz,
which corresponds to a frequency resolution of 0.006 Hz.
We use a standard kernel size of 80 for the median filter in the MED algorithm. The larger the kernel size,
the more noise signal would be captured. Our tests show that a kernel size of 80 is the largest size, which
leads to a safe separation without capturing any energy of the earthquake signal.
**4 Results and Discussion**
**4.1 General Results**
In this section we aim to prove the reliability of our HPS noise reduction algorithm and evaluate the
improvement of the OBS data. We applied the method to synthetic and real teleseismic earthquake data
recorded by the OBS station D10 of the DOCTAR array (e.g., Hannemann et al, 2016, Hannemann et al.,
2017). The synthetics were calculated for a source-receiver epicentral distance of 40° (focal depth: 45 km,
focal mechanism: double couple, source duration: 4 s) by using the full wavefield software qseis (Wang,
1999) and a modified average ak135 velocity model including a water layer (Kennett et al, 1995). The
crustal structure of the velocity model is adapted to the 11.5 km deep oceanic crust in that area and the
water depth is fixed to 5 km. Real oceanic noise of the ZRT components recorded by the station D10 is
added to the corresponding components of the synthetic teleseismic signal. We created synthetics for three
different noise scenarios at the beginning (N1), during (N2) and after (N3) tidal currents (Fig. 3) each with
theoretical SNR of 1–10 between noise and P-onset on pure synthetic Z. Throughout the whole paper the
SNR is defined as root mean square (RMS) of the signal divided by RMS of the noise. For further details of
synthetic data creation see Fig. S2. For the comparison with real data, we selected in total 46 teleseismic
events with Magnitudes Mw >5.6 and epicentral distances of 30–160° (see Fig. S1).  Here only those
events were used, where a P onset could be visually identified. The pre-selection of the events is taken from
Hannemann et al. (2017) and expanded by some events with low magnitudes (see Table S1).  In the
following, we will discuss the improvement of the records by comparing the seismograms and
spectrograms of synthetic data and confirm it with real data. We also verify the improvement for two
seismological applications (teleseismic surface wave group velocity analysis and receiver function
analysis). For some observations, e.g. checking the phase arrival of the teleseismic body waves, we rotated
the arbitrary orientated horizontal components of the real data into the ZRT system. The orientation angles
are taken from the previous study on the DOCTAR array (Hannemann et al., 2016).



Comparing the spectrograms and waveforms of the synthetic example a significant improvement of the
SNR is seen in the HPS processed data set on all components  (e.g. Fig. 3 and Fig. S3–5 for the real data).
The continuous spectral lines of the assumed electronic noise are removed from the data, as well as most of
the spectral lines related to tremor episodes of head-buoy strumming. During the tides, we observe a
reduction of the spectral amplitudes for the tilt noise, as well as for the general background noise (Fig. 3
and Fig. S3–5) on the horizontal components. The results concluded from the spectrograms are confirmed
by the spectra (Fig. 2b), which show the removal of the spectral peaks of the electronic noise (0.05, 0.1,
0.15 Hz) and the tremor episodes (0.5–1 Hz). The amplitudes of the frequencies corresponding to the
teleseismic event and its waveforms are maintained (see Fig. 3).
To quantify the improvements of the method, we calculated the cross-correlation of the teleseismic
waveform, the SNR of the teleseismic body-wave phases and the RMS of the teleseismic waveform before
and after denoising. Because most of the oceanic noise ranges at frequencies below 1 Hz, which is also the
most interested frequency range of the OBS analysis, a 1 Hz low pass filter is applied to the signals before
all result calculations.
We calculated the correlation coefficient for synthetic SO and HPS compared with the synthetic earthquake
signal for different SNR and noise realizations and plotted it in Fig. 4a. The high correlation coefficients for
HPS and synthetic compared with SO and synthetic in all cases demonstrate that HPS denoising preserves
the earthquake signal and doesn't introduce waveform distortion.
For the SNR calculation we used a signal window of 30 s starting from the theoretical onset (direct P on Z
component, direct S on R and T component and Love wave on the T component) and a noise window of 60
s starting 70 s before the theoretical onset. For the Love wave, the SV phase (R component) and P phase (Z
component) the SNR increased significantly (Fig. 4b). For SH phase on T component we observe a few
apparent SNR decreases comparing SO with HPS traces (Fig. 4b). The SNR is calculated on the noise
contaminated SO traces, it hence compares noise with noise contaminated synthetic signal. Because of this
and because we added the synthetics amplified according to SNR for P-arrival on Z component to the real
noise, in a few cases we observe apparent SNRs slightly below 1 for a few SH phases (see Fig. 4b).
The RMS amplitudes of a noise free R component synthetic, SO and HPS signals are estimated over 8
seconds windows with 80% overlap and plotted in Fig. 4c. Comparing the RMS amplitude of the synthetic,
SO and HPS we see that the synthetic and HPS have similar amplitude ranges while SO has a much higher
amplitude. This shows a significant noise reduction in HPS along with preserving the earthquake energy.
As there is some noise remaining after denoising we see some differences in the overall shapes of the RMS
amplitude of the synthetic and HPS (especially after minute 24 which is almost at the end of the energy of
the synthetic signal), however HPS shows peaks on the arrival times of seismic phases of the synthetic
which means that the energy of seismic phases is preserved after denoising. The minor changes in seismic
phase shapes of the synthetic and HPS is also due to remaining noise. The seismograms and spectrograms
related to this example are presented in Fig. 3. Figure 4d shows a comparison of RMS amplitude of the
original noise in SO (blue curve), the remaining noise in HPS after denoising (red curve) and the synthetic





earthquake (green curve) signals. Besides a high noise reduction in HPS, the plot shows that the remaining
noise is independent from the pattern of the synthetic earthquake, which confirms that denoising process
doesn't affect the earthquake energy in the HPS signal.
**4.2 Applications**
By applying our HPS noise reduction algorithm, we aim to improve seismological analysis, especially
those involving the analysis of teleseismic body and surface waves. Valuable constraints of the Earth's
structure in oceanic regions can be taken from the analysis of the SH-wavefield like Love-waves, which are
not influenced by the water column, but often cannot be analyzed due to strong noise on the horizontal
components. SV waves are also often masked by noise, but are for instance important for tomography
studies or S and SKS shear wave splitting analysis (e.g. Silver and Chan, 1991). Other techniques using the
SV-wavefield like the Z/R ratio of the teleseismic Rayleigh waves (Tanimoto & Rivera, 2008), or receiver
functions (RF) (Langston, 1979) also rely on clear radial component readings.  In the following we will
show the improvement which was achieved for the SH arrivals and for the group velocity analysis of
teleseismic Rayleigh- and Love waves, as well as for the receiver function analysis.
**4.2.1 SH-waves**
Since SH-waves are weak in energy and displayed on the noise-contaminated transversal horizontal
component (T), they are sparsely observable on OBS data and mostly disappeared behind the high noise
level. However, on the HPS processed data we see an improvement of the SNR on the T-component (see
Fig. 4b). In many cases the SH-phase is clearly identifiable on the HPS T-component (see Fig. 3d for a
synthetic data and Fig. S6 for a real data example).
**4.2.2 Surface waves**
Rayleigh waves in deep oceanic domains are strongly influenced by the water column, because most of the
wave energy is traveling in the water. This poses a problem, if the water depth changes along the travel
path. Love waves are not influenced by the water column but are recorded only on horizontal components
and their recordings on OBS systems are therefore more disturbed by strong noise sources like tilt inducing
tidal currents. To test the performance of the HPS noise reduction algorithm in the long period range, we
performed a measurement of group velocities of Love and Rayleigh waves with the Multiple Filter
Technique (MFT) (Dziewonski et al., 1969). Group velocity curves are for instance used as input data for
tomographic studies to reveal the 3D structure of the lithosphere and upper mantle. Figure 5 shows group
velocity curves for the synthetic Love wavetrain for the three noise situations N1-N3. For the MFT analysis
we used the software mft96 (Herrmann, 2013). The unfiltered seismograms in the top panels (Fig. 5a–c)
correspond to the P-wave SNR = 1 scenario. In all three cases the clarity of the dispersion curve is greatly
enhanced in the images resulting from the HPS processed traces (Fig. 5e–g) in comparison to the noise free
image (Fig. 5d). Also the seismogram traces improved greatly. The dispersion maps show that also noise



energy in the range of the signal frequencies is removed successfully in the frequency range 0.05 to 0.2 Hz.
Longer signal periods which are weakly visible in the noise-free image (Fig. 5d) can not be recovered. The
corresponding results for the Rayleigh wavetrain on the radial component are shown in Fig. S7. For the N3
case here also longer periods down to 40 s can be successfully denoised.
For an evaluation of the HPS denoising technique on real surface wave data we selected 23 events with
magnitudes larger than Mw 6.0 in the distance range between 47.5° and 159.6° and added one event with
Mw= 5.6 at a distance of 37.9° (see Fig. S1). Figure S8 shows seismograms and MFT analysis examples
for three events with different magnitudes and in different distances. The resulting group velocity
dispersion curves for all 24 events for the original and processed data are shown in Fig. S9. For all
components we find that the improved signal to noise ratio of the processed data allows the analysis of
more events and of a broader period range than in the original data.

**4.2.3 Receiver Functions**
Receiver functions have been proven to be a valuable tool to observe the Earth's structure using teleseismic
events (e.g., Langston, 1979, Ammon et al., 1995, Kind et al., 1995; Rondenay, 2009). Separating the
source site from the receiver site by deconvolution allows to estimate the Earth's structure beneath the
station. Here, we compare the receiver functions calculated from the synthetic examples and from real data
before and after denoising (Fig. 6). The synthetics used for the receiver function calculation are pure
synthetic signals contaminated by real noise (N1, N2, N3). On the synthetics, the SNR for P ranges
between 1–10 (for detailed description of the synthetic creation, see Sect. 4.1, Fig. 3 and S2). Receiver
function analysis and the observation of the Earth's structure beneath the DOCTAR array was already
calculated by Hannemann et al. (2017). Here, we don't aim to estimate the crustal and mantle structures,
instead we aim to compare the P-receiver Functions of the radial component calculated from the original
synthetic and real data (SO R-RF) with receiver functions of the radial component from the HPS processed
synthetic and real data (HPS R-RF). To calculate the receiver functions, we applied the iterative
deconvolution in the time domain (Ligorría & Ammon, 1999). We corrected the data for the Ps-phase,
quality controlled (e.g. P-onset at 0 s on Z of HPS R-RF), stacked and low-pass filtered the synthetic data at
2 Hz and bandpass filtered the traces between 0.05–0.5 Hz for the real data with a zero-phase Butterworth
filter. For both synthetic and real receiver function, the noise level strongly decreased and we observe a
significant decrease in variance on the HPS traces compared to the SO traces (Fig. 6).
Our result shows that determination of the crustal- and mantle-phases is more reliable on the HPS R-RF
stack than on the SO R-RF stack for both synthetic and real data (Fig. 6). We observe more distinct Ps-
phase arrivals on the HPS R-RF than on the SO R-RF stack. The Ps-phases are caused by the P-to-S
conversion at the Mohorovičić-, 410-km and 660-km discontinuity (hereafter referred to as Moho, 410, and
660, respectively; e.g. Deuss, 2009). For the synthetic example, we expect the P-to-s conversion at the
Moho at depths of 11.5 km to arrive at 0.8s, which is better resolved in the synthetic HPS R-RF than in the
synthetic SO R-RF, same for it's multiple ($P_MSPp$) and the water multiples every 6.5s ($M_{WATER}$, Fig. 6a).



Assuming ak135 velocities we would expect the $P_{410}s$-phase (Ps conversion at the 410) to arrive at around
43 s and the $P_{660}s$-phase (Ps conversion at the 660) at around 66.8 s delayed to the direct P-arrival (see Fig.
6 a & b).

Instead of a rather weak peak on the SO-R-RF real data stack we observe a strong peak at around 43 s, with
a good SNR on the HPS R-RF stack, indicating the sharp velocity contrast at the 410 (Fig. 6b).  Comparing
the SO-RF and the HPS-RF real data stacks, the amplitudes of the $P_{660}s$-phase decreased and became a
broader peak, which we would expect from a conversion at a more gradual velocity contrast at the 660 (Fig.
6b). These results are in line with the analysis of the crustal and mantle structure beneath the DOCTAR
array presented by Hannemann et al. (2017). The negative phase (X1 in Fig. 6b) arriving at around 5 s is
stronger on the HPS-R-RF real data stack than on the SO-R-RF real data stack and might either indicate the
PpSs multiple of the Ps-phase at the Moho, or the direct P-to-s conversion at the LAB.  On the HPS-R-RF
real data stack we observe a strong positive phase (X2) arriving at 12 s (Fig. 6b). This phase has not been
identified by Hannemann et al. (2017) and a detailed analysis of its origin is beyond the scope of this study,
but it might be related to the water multiples.
In general, receiver functions of OBS data are difficult to analyze and although the SNR of the HPS
processed data has been improved, the analysis of the real data is still difficult, especially compared to
receiver functions from land stations.

**5 Conclusions**

In this work we have developed a method to separate the signals of teleseismic earthquakes from other
signals in the OBS data resulted in noise reduction of OBS data. Our method is a combination of two HPS
algorithms from the field of MIR to separate harmonic and percussive components of an OBS data.
Earthquake signals as percussive components are separated from noise signals as harmonic components.
The noise signal is reconstructed using the phase information of the original signal. Subtracting the noise
signal from the original signal derives the noise-reduced signal. We discussed the motivation of using a two
step HPS approach, that results in a clean noise-reduced signal where the teleseismic broadband earthquake
waveforms are preserved with their whole frequency content. We also discussed the type of noise signals,
which are eligible for our noise reduction algorithm that contains most of the OBS noise energy.
The extracted noise signal contains some different signals where each can be derived by applying a band
pass filter to the extracted noise signal in a proper frequency band. The derived signal may be used in
researches related to that signal. For example the microseism signal can be extracted and used for
investigation of the source generation area of microseisms.

The comparison of original and HPS noise reduced synthetic signals shows how significantly the SNR has
improved after applying our method (Fig. 4b). However, the apparent SNR improvement highly depends on



the noise type characteristics. For Noise type 1 (N1), it seems like there is no SNR improvement on the T-
component (Fig. 4b, the second panel). N1 is taken from the beginning of the tidal current event, where we
have a considerable time dependent change in noise-frequencies. In this example, the noise has similar
frequencies with the signal. The visual inspection of the corresponding trace indicates a clear improvement
of the waveform for SH-wave on T component, even though the SNR shows no improvement. The results
from the cross-correlation (Fig. 4a), confirm the improvement and preservation of the waveform. Since we
are focusing on the preservation of the waveform and the SNR comparison highly depends on the noise
situation, the SNR should not solely be used to evaluate the improvement by HPS noise reduction method.
Even if the SNR is not improving in a few cases, the analysis of the cross-correlation, RMS and the pure
waveforms, verify the improvement of the traces by the HPS noise reduction method. From our analysis of
the broadband seismograms, we find out that the improvement is significant and may allow a broader and
more reliable analysis of teleseismic earthquake data. Applications like the receiver function technique and
SH-wave and Love wave analysis are considerably improved after applying the HPS noise reduction
algorithm. For the receiver function, we could observe a more distinct phase for the P-to-S conversion at
the Moho for the synthetic case and at the 410 km discontinuity for the real data. Group velocity analysis of
teleseismic surface wavetrains showed that application of the HPS noise reduction technique allows to
analyze more events and to analyze them in a broader frequency range. Especially more and wider Love
wave dispersion curves could be recovered. The noise reduction algorithm improves the horizontal
components significantly, which allows the OBS community to apply a broader range of seismological
methodologies, including the horizontal components, to the OBS-data.
In conclusion, the presented method is a powerful algorithm for separation and extraction of different
signals from OBS data and has especially application in noise reduction of OBS signals.
**Code and data availability**
The Python code related to the proposed method along with an example of real data is freely available from
https://github.com/ZahraZali/NoiseCut. A Jupyter notebook with all the Python codes and parameters relat-
ed to the proposed method is available as an electronic supplement. The sea floor seismological data were
archived by Alfred Wegener Institute (AWI), Helmholtz Centre for Polar Research, Bremerhaven, Germa-
ny, and are available upon request. The supplementary material related to this article contains list of all
earthquakes used in this study and a map showing their location. The illustrations of the semi-synthetic data
generation are presented in the supplementary material as well. An example of three components seismo-
gram and spectrogram before and after applying HPS noise reduction algorithm to real data, Rayleigh wave
group velocity analysis for a synthetic example, MFT analysis for three real events and group velocity
curves for some real events are also presented through figures in the supplementary material.

**Author contribution**



Z.Z. developed the algorithm and designed the study. T.R. created the synthetics data, conducted the
synthetic tests, and measured the receiver functions. F.K. conducted the group velocity analysis. Z.Z., T.R.,
and F.K. evaluated the results. Z.Z. and T.R. wrote the initial draft. All authors wrote the final manuscript
and discussed the results.

**Competing interests**
The authors declare that they have no conflict of interest.
**Acknowledgments**
Zahra Zali is grateful for the support by the German Academic Exchange Service (DAAD) through the
Graduate School Scholarship Programme under reference number 91721165. This work was also supported
by the German Research Foundation (DFG MU 2686/13-1, SCHE 280/20-1) and the Daimler Benz
Foundation (32-02/18). The sea floor seismological data were archived by Alfred Wegener Institute (AWI),
Helmholtz Centre for Polar Research, Bremerhaven, Germany, and are available upon request. We
acknowledge the DEutscher Geräte-Pool für Amphibische Seismologie (DEPAS) pool (Alfred-Wegener-
Institut Helmholtz-Zentrum für Polar- und Meeresforschung et al., 2017) that is currently the largest
European OBS pool. We acknowledge Sebastian Heimann for helping in packaging of the code related to
the method. For building our method, we used Librosa, a Python package for audio and music signal
processing (McFee et al., 2020). The data processing was done using obspy (Beyreuther et al., 2010) and
pyrocko (Heimann et al., 2017); the receiver functions were calculated using the rf-package (Eulenfeld,
573   2020).

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

Notes Measured as Sound Pressure and Particle Velocity from Ocean-Bottom Seismometers in the North

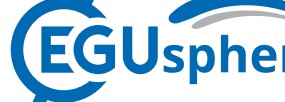

Atlantic. Journal of Marine Science and Engineering, 9(6), 646, https://doi.org/10.3390/jmse9060646,
700    2021.

Pillet, R., Deschamps, A., Legrand, D., Virieux, J., Béthoux, N., & Yates, B.: Interpretation of broadband
ocean-bottom seismometer horizontal data seismic background noise, B. Seismol. Soc. Am., 99(2B), 1333-
1342, https://doi.org/10.1785/0120080123, 2009.
Rafii, Z., Liutkus, A., & Pardo, B.: REPET for background/foreground separation in audio. In Blind Source
Separation (pp. 395-411). Springer, Berlin, Heidelberg, 2014.
Rafii, Z., Liutkus, A., Stöter, F. R., Mimilakis, S. I., FitzGerald, D., & Pardo, B.: An overview of lead and
accompaniment separation in music, IEEE/ACM Transactions on Audio, Speech, and Language
Processing, 26(8), 1307-1335. https://doi.org/10.1109/TASLP.2018.2825440, 2018.
Rafii, Z., & Pardo, B.: Music/Voice Separation Using the Similarity Matrix, Proc. ISMIR, pp. 583-588,
714    2012.

Ramakrushana Reddy, T., Dewangan, P., Arya, L., Singha, P., & Kamesh Raju, K. A.: Tidal triggering of
the harmonic noise in ocean-bottom seismometers. Seismol. Res. Lett., 91(2A), 803-813,
https://doi.org/10.1785/0220190080, 2020.
Rondenay, S.: Upper mantle imaging with array recordings of converted and scattered teleseismic waves.
Surv. Geophys., 30(4), 377-405, https://doi.org/10.1007/s10712-009-9071-5, 2009.
Romanowicz, B., Stakes, D., Montagner, J. P., Tarits, P., Uhrhammer, R., Begnaud, M., Stutzmann, E.,
Pasyanos, M., Karczewski, J.F., Etchemendy, S. and Neuhauser, D.: MOISE: A pilot experiment towards
long term sea-floor geophysical observatories, Earth, planets and space, 50(11), 927-937, 1998.
Schlindwein, V., Krüger, F., Schmidt-Aursch, M.: Project KNIPAS: DEPAS ocean-bottom seismometer
operations in the Greenland Sea in 2016-2017, Alfred Wegener Institute, Helmholtz Centre for Polar and
Marine Research, Bremerhaven, PANGAEA, https://doi.org/10.1594/PANGAEA.896635, 2018.
Schlindwein, V., Wassermann, J., & Scherbaum, F.: Spectral analysis of harmonic tremor signals at Mt.
Semeru volcano, Indonesia, Geophys. Res. Lett., 22(13), 1685-1688. https://doi.org/10.1029/95GL01433,
733    1995.





Skop, R. A., & Griffin, O. M.: On a theory for the vortex-excited oscillations of flexible cylindrical
structures. Journal of Sound and Vibration, 41(3), 263-274, https://doi.org/10.1016/S0022-460X(75)80173-
737 8, 1975.


Silver, P.G. & Chan, W.W.: Shear wave splitting and subcontinental mantle deformation. J. Geophys. Res.,
740 96, 16429-16454, 1991.


Snodgrass, F. E., Hasselmann, K. F., Miller, G. R., Munk, W. H., & Powers, W. H.: Propagation of ocean
swell across the Pacific, Philosophical Transactions of the Royal Society of London. Series A,
Mathematical and Physical Sciences, 259(1103), 431-497, https://doi.org/10.1098/rsta.1966.0022, 1996.

Stähler, S. C., Schmidt-Aursch, M. C., Hein, G., & Mars, R.: A self-noise model for the German DEPAS
OBS pool. Seismol. Res. Lett., 89(5), 1838-1845, https://doi.org/10.1785/0220180056, 2018.

Tanimoto, T., Rivera, L.: The ZH ratio method for long-period seismic data: sensitivity kernels and
observational techniques. Geophys. J. Int., 172(1), 187-198, https://doi.org/10.1111/j.1365-
246X.2007.03609.x, 2008.

Titze, I. R.: Nonlinear source–filter coupling in phonation: Theory, The Journal of the Acoustical Society
of America, 123(4), 1902-1915, https://doi.org/10.1121/1.2832337, 2008.

Vaseghi, S. V.: Advanced signal processing and digital noise reduction. Vieweg + Teubner Verlag, 1996.

Wang, R.: A simple orthonormalization method for stable and efficient computation of Green's
functions, B. Seismol. Soc. Am., 89(3), 733-741, https://doi.org/10.1785/BSSA0890030733, 1999.

Webb, S. C.: Broadband seismology and noise under the ocean, Rev. Geophys., 36(1), 105-142,
https://doi.org/10.1029/97RG02287, 1998.

Webb, S. C., Zhang, X., & Crawford, W.: Infragravity waves in the deep ocean, J. Geophys. Res.-:
Oceans, 96(C2), 2723-2736, https://doi.org/10.1029/90JC02212, 1991.

Zali, Z., Ohrnberger, M., Scherbaum, F., Cotton, F., & Eibl, E. P.: Volcanic Tremor Extraction and
Earthquake Detection Using Music Information Retrieval Algorithms, Seismol. Res. Lett., 92(6), 3668-
3681, https://doi.org/10.1785/0220210016, 2021.



Zhu, W., Mousavi, S. M., & Beroza, G. C.: Seismic signal denoising and decomposition using deep neural
networks. IEEE T. Geosci. Remote., 57(11), 9476-9488. https://doi.org/10.1109/TGRS.2019.2926772,
773    2019.


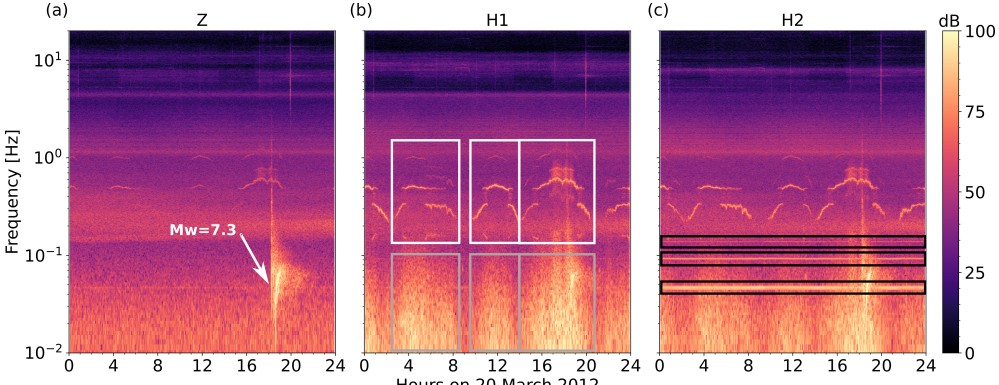

**Figure 1: Spectrogram of an one-day OBS signal shows ocean bottom noise on Z (a), H1 (b) and H2 (c)**
**components. The data was recorded by the station D10 of the DOCTAR array with a sampling frequency of 100**
**Hz. The spectrogram were calculated using a window length of $2^{16}$ sample and an overlap of 75%. The signal of**
**an earthquake (Mw=7.3) on 20.3.2012 at around 18:00 at the station D10 is shown in (a). The tidal cycle of the**
**current-induced noise is clearly visible during the high tilt noise episodes (grey box in b). The white box in (b)**
**highlights the tremor episodes caused by the head buoy strumming. On H2 (c) we see an instrument-related,**
**presumably electronic noise (black boxes). The high energy of the secondary microseism band at around 0.2 Hz**
**is visible on all components.**

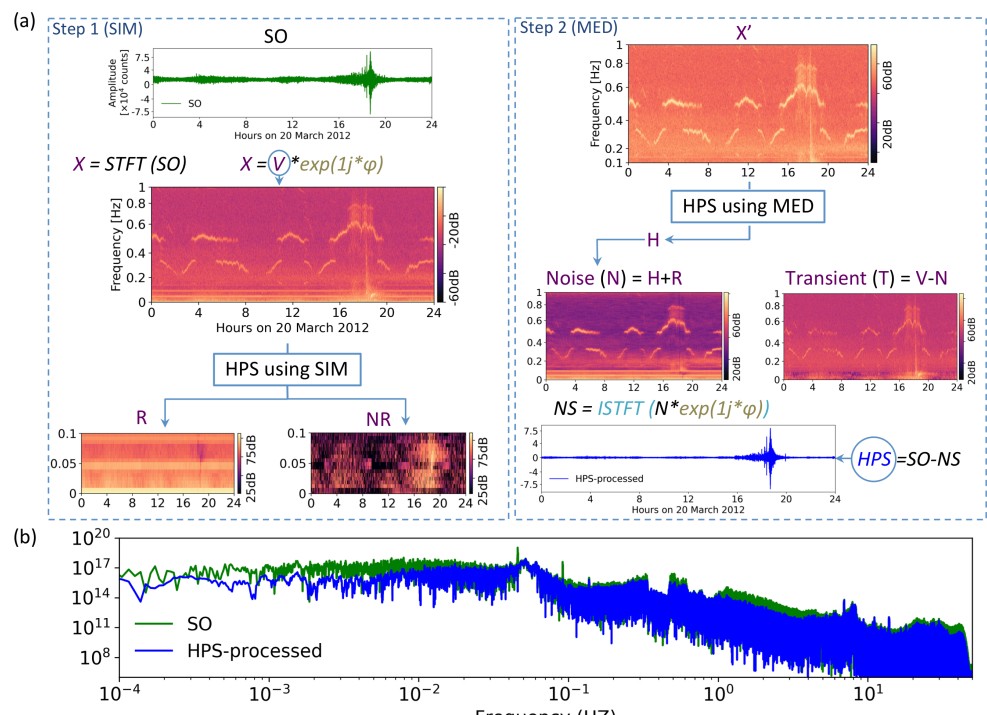

**Figure 2: Method flowchart (a) Illustration of the processing steps with a real data example. Left panel shows**
**the first step of the method where using the similarity matrix (SIM) in the frequency range below 0.1 Hz and**
**above 1 Hz, we divide the spectrogram of the signal into two spectrograms of repeating (R) and non-repeating**
**(NR) patterns. Right panel shows the second step of the method where we apply a median filter (MED) to the**
**frequency range of 0.1 to 1 Hz in order to remove noises from this frequency range. As the interested frequency**
**range for OBS signals is below 1 Hz, the spectrograms show only this frequency range. Finally the noise**
**spectrogram (N) is created by summing the separated noises derived from two steps and the noise signal (NS) is**
**derived using ISTFT. We obtain the noise reduced signal (HPS) by subtracting the NS from the input OBS**
**signal (SO). STFT, short time Fourier transform. HPS, harmonic-percussive separation. SIM, similarity matrix.**
**MED, median filtering. ISTFT, Inverse Short Time Fourier Transform. (b) Spectrum of the original signal (SO)**
**and the HPS noise reduced signal.**



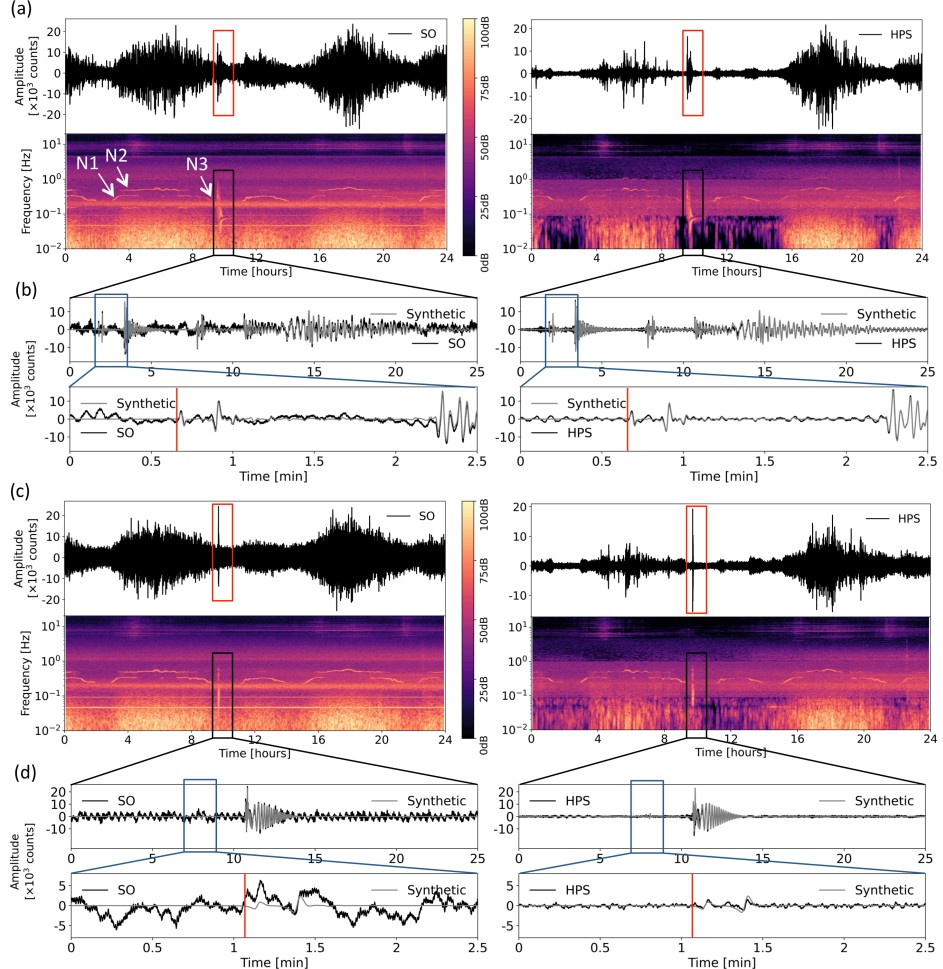

**Figure 3: Comparison of the synthetic seismograms and spectrograms of the original signal SO and the HPS noise reduced signal on the R and T components for a synthetic signal with SNR= 1.5 before denoising (SNR is defined as RMS of the signal divided by RMS of the noise). (a) & (c) Show one day seismograms and spectrograms for R and T components, respectively. The spectrograms clearly show the reduced noise level on the HPS signal. Squares show the earthquake section. The arrows in (a) show three noise situations (N1-N3). (b) & (d) Show seismograms of the earthquake section on SO and HPS signals, with detailed view of the P-arrival (on component R in subfigure b) and SH-arrival (on component T in subfigure d). The whole amplitude and the phase information of the synthetic earthquake are preserved in the HPS signal but it's very less noisy compared with SO. Red lines show P-arrivals in (b) and SH-arrival in (d).**



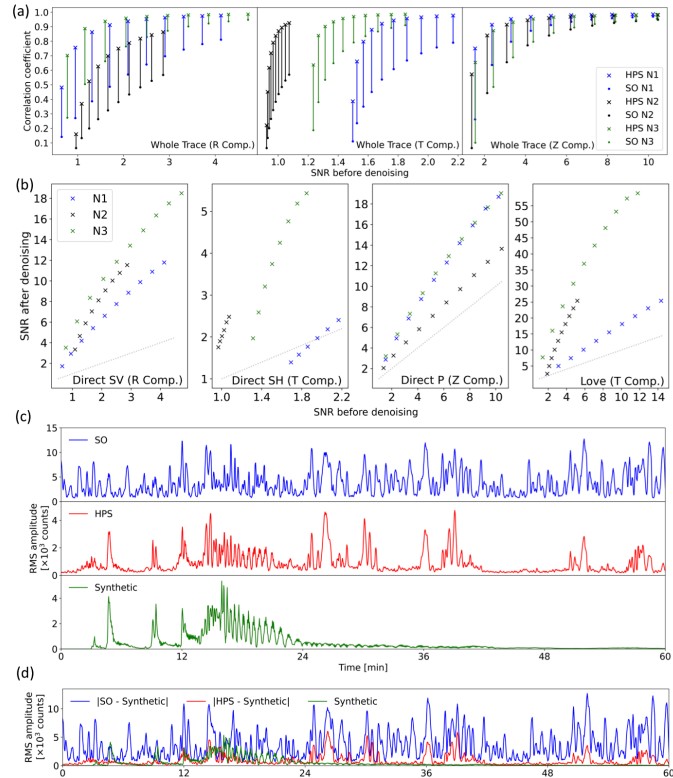

**Figure 4: Comparison of the synthetic SO and HPS signals (both are lowpass filtered at 1Hz). (a) Correlation coefficients (for the whole trace) for different SNRs and 3 realistic noise realizations for Z, R and T components (Component is abbreviated as Comp.). (b) Improvement of SNR for direct body wave phases and the Love wave. We see significant improvement in both correlation coefficient and SNR for all the realizations after denoising. The gray dotted lines in (b) mark the line with gradient 1 (no improvement of SNR). (c) Comparison of the root mean square (RMS) amplitude of one example of the SO, HPS and synthetic earthquake signals. The HPS signal has significantly lower energy compared with SO due to noise reduction, but has almost similar energy compared with the synthetic earthquake which shows the energy of the earthquakes and all the phase arrivals are well preserved during the denoising process. This signal is the same example shown in Fig. 3 (R component, SNR= 1.5 before denoising). (d) The RMS of the original noise (blue trace: |SO - Synthetic|) and the remained noise after denoising (red trace: |HPS- Synthetic|) compared to the synthetic earthquake signal. A high noise reduction is seen in the red trace compared with the blue one and the remained noise has an inconsistent pattern compared to the synthetic earthquake that confirms denoising process doesn't modify the earthquake energy and its phase arrivals.**

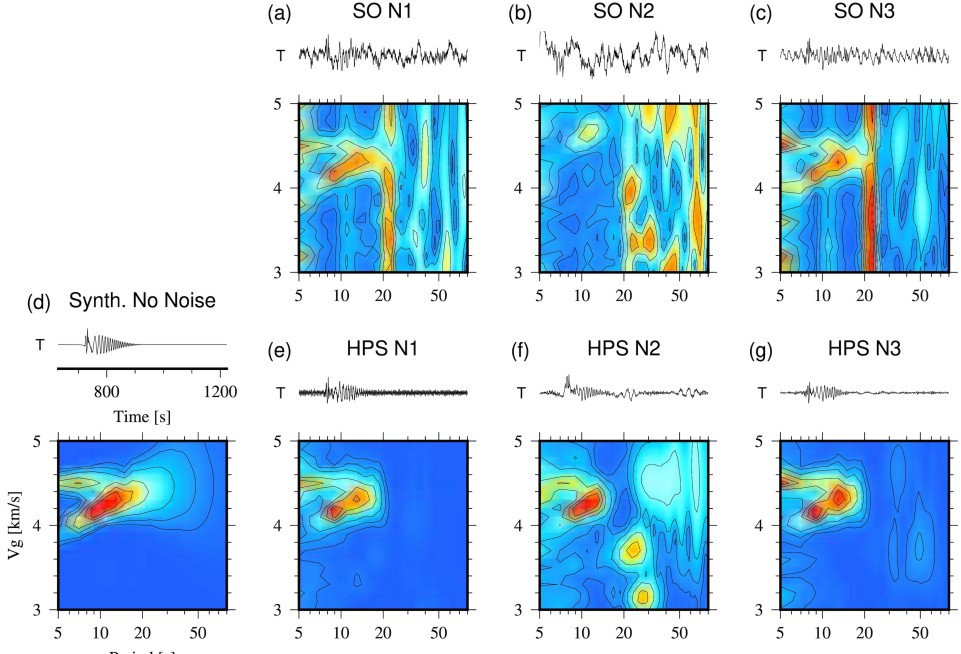

**Figure 5: Love wave group velocity analysis for unfiltered and HPS processed synthetic Love wavetrains contaminated by three real world OBS noise signals (noise signals N1-N3, station D10, DOCTAR experiment, see Sect. 2 for more details). (a)–(c): Lower panels: Unfiltered synthetic signal (SO) MFT analysis results. Top panels: seismogram time windows corresponding to the range of group velocities shown on the y-axis. (d) Noise free synthetic case. (e)–(g): HPS processed input traces for noise situations N1-N3 (lower panel: MFT analysis result, top panel: HPS processed seismogram).**



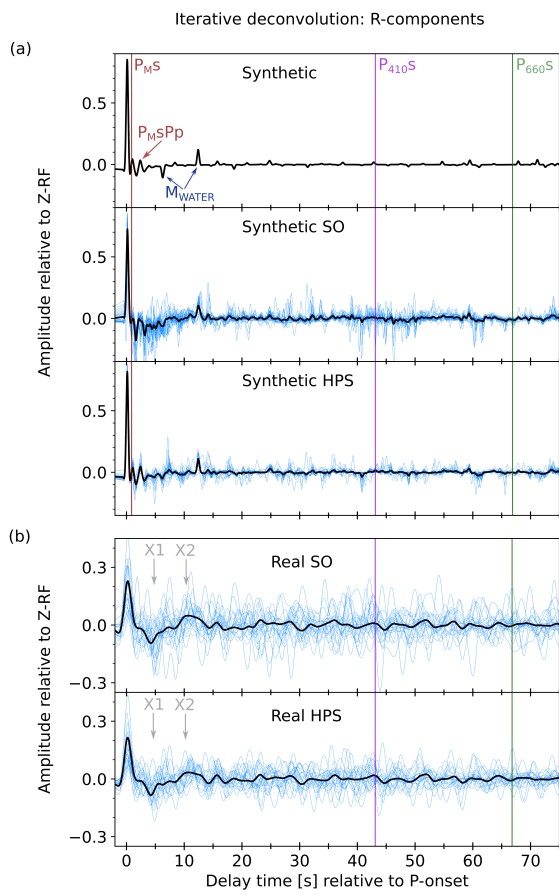

**Figure 6: R-receiver function comparison of synthetic and real data examples. (a) Comparison of the synthetic data examples, lowpass filtered at 2 Hz. The pure synthetic R-RF is shown in the uppermost panel, followed by the synthetic SO and the synthetic HPS R-RFs. The black lines show the summed individual R-RFs (blue waveforms). The theoretical onset times for this specific model are marked. Red line: Ps-arrival of the Moho (P$_M$S) and its multiple (P$_M$SPp), violet line: Ps arrival of the 410 (P$_{410}$s), green line: Ps-arrival of the 660 (P$_{660}$s), dark-blue arrows: Multiples in the watercolumn of 4.9 km (M$_{WATER}$), repetitive every 6.5s. (b) Comparison of the real data, bandpass filtered at 0.05–0.5 Hz. The upper panel shows the R-RFs of the real SO traces and the lowermost panel the R-RFS of the real HPS traces. The individual traces (blue) are shown as stack (black line) and the theoretical onset times based on the average ak135 velocity model are shown as violet line (P$_{410}$s) and green line (P$_{660}$s). The origin of the phases X1 and X2 (grey) remain unclarified, since their interpretation is beyond the scope of this study.**