# Peer review of "components using harmonic-percussive separation"

_EGUsphere, 2022_

## Author Response (AR1)

**Comments from Referee 1:**

*RC: Generally, I would have liked to see this method applied to a larger dataset. DOCTAR only covered a small area. I would expect that the signals and noise throughout the network are similar to each other. Noise in other parts of the Atlantic Ocean, the Pacific Ocean or the Indian Ocean might show other challenges. It would have been nice to see the method applied to other OBS experiments.*

AC: We appreciate your comment on applying the HPS denoising algorithm to other OBS data with different noise sources. We have already used the presented algorithm to denoise data from the "KNIPAS" project (Schlindwein et al., 2018) and have found significant improvement when calculating receiver functions (Rein et al., EGU22, manuscript in preparation). We are therefore quite confident that the algorithm is able to significantly reduce noise from different sources as far as they are long lasting narrowband signals, which is the signature of many important OBS noise signals. Attached we show one denoising example from the KNIPAS data set being one part of another paper under preparation. We have restrained from including more examples for different data sets into the submitted manuscript to avoid the need for presenting the different experiments and noise conditions that would inflate the size of the paper significantly.

*RC: L.148 - 154: You are mentioning the head buoy as a source of harmonic noise, but what about the flag? Can it also play a role or is the strumming of the head buoy overshadowing the flag signal?*

AC: According to Essing et al., (2021), the noise source of the flagpole is most likely depending on the orientation of the OBS instrument, since it is fixed directly on the frame of the OBS. Essing et al. (2021) have analyzed the tremor noise sources in detail and did not observe any dependency of the tremor signal on OBS orientation. Therefore the head buoy is most likely the predominant noise source for the tremor events.

*RC: L. 351 - 353: You are using qseis for generating synthetic seismograms. Were Source Time Functions (STF) used for synthetics? It is a minor point (and no action is required). There is probably no considerable difference between the tests, but a more realistic STF might even improve some of the cross-correlations.*

AC: For generating the synthetics with qseis, we used a normalized square half sinus as STF with duration of 2 s. A more realistic STF would change the source spectrum but not its wideband (transient) characteristic, which is the basis of the separation of the earthquake signal from long duration narrowband noises using our HPS denoising algorithm.

*RC: L. 356-358: Your noise sources were picked at the beginning (N1), during (N2) and after (N3) tidal currents. How do you ensure the signals are not "contaminated" with other noise sources, such as storms or ships? Would that even matter for the analysis?*

AC: We ensured that these noise scenarios are not contaminated with other noise sources acting at frequencies below 1 Hz, however even if other noise sources would exist, the analysis is independent from the type of the noises. As stated above, the

denoising algorithm will remove mostly the long lasting narrowband noise type, which is typical for OBS recordings.

*RC: L.542-551: It would be nice to mention how efficient the algorithm is.*

AC: Thanks for the remark. We added the below sentence to mention the efficiency of the algorithm.
*An example of one day OBS signal with a sampling frequency of 100 Hz is presented in the GitHub page. The average computation time for this example is about 7 minutes on a PC with an Intel core i7 (six-core) processor of 2.2 GHz and 16 GB of RAM.*

*RC: Figure 1: Why don't you show the hydrophone channel? I think it would be nice to see it as a comparison.*

AC: We haven't shown the hydrophone channel since we feel it does not provide much additional information here. The main purpose of this study is to reduce noise from horizontal components and make use of consistent patterns determined from the spectrograms. Other algorithms (Crawford and Webb, 2000; Bell, et al., 2015) designed for denoising only the vertical component of OBS recordings certainly make use of the hydrophone channel. However, in this study, the hydrophone is not needed for the denoising of the vertical and horizontal components. Therefore we have decided to show only those components, which are used.

*RC: Figure 2: This figure is a bit confusing because not all the acronyms are in the caption (e.g. X', H, R…). They are defined in the text of the manuscript but it would be nice for completeness to have everything in the caption.*

AC: Thanks for the comment. The caption is modified and the information is added.

*RC: Figure 3: Here, it looks like the N3 noise source is close to the earthquake (or the arrow is even pointing at the earthquake). Did you ensure that the noise, which was added to the synthetics, was earthquake free?*

AC: The earthquake shown in Figure 3 is the synthetic earthquake, which was added to the real noise data at the position of noise type N3. In this Figure we show the improvement of the HPS noise reduction algorithm on the R and T component, using the illustrated synthetic earthquake at N3 as an example. However, for N1-N3, we have ensured to only add earthquake-free noise data to the synthetic earthquake. We have changed the illustration of arrows in figure 3 to clearly point the noise and not the earthquake signal.

**Comments from Referee 2:**

*RC: The authors present the parameters used in their algorithm in a haphazard manner and often before they have explained why these parameters are needed. For example, on line 214 they indicate that they divide the frequency content of the signal into two ranges and they give the values of the ranges, but they don't explain why this division is needed and how they choose the ranges until lines 271-272 and 301-303.*

*The authors should reorganize the text so that the need for parameters and the criteria used to select these parameters is explained from the beginning. This will allow others to more easily understand and profit from their algorithm. I also recommend that the authors make a table of these parameters.*

**AC**: We appreciate your comment on explaining the parameters of the algorithm in a more structured manner so that the reader can easily understand them. We reorganized the manuscript and add more information at the beginning. However we keep the detailed explanation about the reason of two frequency ranges in the method section after we explain SIM. The reason is that the reader needs to know how SIM works to clearly understand it. Now we mention this at the beginning of the method section as well so the reader knows that he/she will have a better understanding of this until the end of the section. We created a table, which contains all the parameters, which we used in our study. The explanation about the parameters is presented in the text.

*RC: The figure captions often contain expository text that should be in the main text and lack specific information about the figures themselves (see below).*

**AC**: We modified figure captions; removed the unnecessary information, which exist in the text, and added more information about the figure itself.

*RC: There are some language issues that do not prevent understanding but slow down reading, including extraneous or missing "the"s and overuse of "in order to". Below is an incomplete list of more complicated examples, with suggested corrections:*
  *- L313: "part of that is still remained" -> "part of it remains"*
  *- L314: "The signals ... that don't ... in the spectrogram are difficult to be captured by our HPS algorithm so part..."*
  *-> ""Signals ... , which don't ... in the spectrogram, are difficult to capture using our HPS algorithm, so part...""*

**AC**: We applied the suggested corrections as well as some further language modifications.

*RC: L53-59: The details of microseism noise aren't relevant to the performed analysis/results.*

**AC**: We shortened the text and removed details of microseism noise.

*RC: L82: The IG wave signal used by Crawford and Webb was not recorded by a hydrophone, but by a differential pressure gauge. Differential pressure gauges, nano-precision bottom pressure recorders or broadband hydrophones can be used to measure the IG wave signal, though I'm not sure if broadband hydrphones are sensitive enough below their corner frequency.*
**AC**: Thanks for this remark. We replaced "hydrophone data" by "differential pressure gauges".

*RC: L136-141: These details of the LOBSTER OBSs development aren't relevant to the method or the data presented.*

**AC**: We removed this part from the manuscript.

*RC*: *L175-176: Repeats previous lines.*

**AC**: The sentence is removed.

*RC*: *L180-192: In this description of the MED algorithm, it is not clear which bits are information about median filters and which are intrinsic to the specified algorithm.*

**AC**: This is true. However in this subsection we only described the MED itself as the subsection title shows as well. In the following subsection 3.4 we described how we used MED and SIM in our algorithm.

*RC*: *L210-212: The statement that most OBS noises are narrow-band is false for tilt, microseism and compliance noise.*

**AC**: We modified the sentence. OBS noise signals are more narrow-band compared to earthquake signals. The different frequency characteristic of earthquakes and the OBS noise is an important feature that makes HPS suitable for separating them.

*RC*: *L301-303: This should be explained in the algorithm section*
*& L298-300: This should be written more clearly*
*& L 214-215: Why 0.1 to 1 Hz? Why two ranges? This is only explained ~60 lines later.*

**AC**: We reorganized and modified the text so some information has been moved to the beginning part of the algorithm description. However, for the reader it is necessary to know how the SIM works to clearly understand the reason for dividing the frequency range into two parts. At the beginning of the Sect. 3.4 we mention the reason for this division briefly. Later we explain it in more detail after the reader knows how the algorithm works. The ranges are now shown in the table parameters as well.

*RC*: *L 235: Why 2%? Is this a parameter you set? Or an observation of some separation in S-values?*

**AC**: We use a threshold for picking the highest similarity. We choose the upper 2% of the time frames with highest S values as the similar frames. We modified the sentence and added the term "the upper" to make it clear.

*RC*: *L306: "will be separated in the": is this another step? or the output of this step?*

**AC**: This is the output. We modified the sentence to make it clear.

*RC*: *L320: Does the phase component have a name? (Since the amplitude is name "V").*

**AC**: As it is explained in the text, equation 1 separates X into its amplitude (V) and phase components (which by looking to the equation it is clear that the phase component is: $exp(1j * \varphi)$) where $\varphi$ is the phase of X (written in the text). Naming amplitude component as V helps to better understand figure 2, but it is not needed to write a specific name for the phase component.

*RC*: *Eq 7: Use the same emphasis in the equation as in the text (N and N' are bold in the text, but italicized in the equation)*

**AC**: Within the whole manuscript, we used bold for the variables in the text and used italic for the equations.

*RC: L337: the window length and overlap should be in the parameter table and the chosen values explained.*

**AC**: In the sections 3.6 it is explained that for extracting narrow-band signals a high frequency resolution is needed in the spectral domain. So it is clear that a long time window should be used for the STFT. We mentioned our recommendation, however one can use other sizes for the FFT window as far as it is large enough to create sufficiently high resolution in the frequency domain to be able to capture the narrow band nature of noise signals. We have mentioned our choice both in the text and table for allowing the reader to reproduce the results.

*RC: L339: is the frequency resolution relevant?*

**AC**: Yes, this is relevant since we want to emphasize that the algorithm doesn't destroy the low frequency content and that the corresponding waveforms are well preserved/reconstructed after denoising.

*RC: L341: Should your choice of a kernel size of 80 (parameter table!) be used by other users, or should they run their own tests?*

**AC**: We added more information about the kernel size and how to choose it. 80 is our recommended size but users may want to capture more noise at the cost of probable minimal waveform distortion, so they can choose a larger size. Using the exact recommended size is not critical for the algorithm, but the user should be able to understand the effect of this parameter and tune it based on the application. So we provide the information here and explain how to choose the kernel size. We also present the chosen value in the table.

*RC: L354: 11.5 km deep oceanic crust: Are there 11.5 km of sediments over this crust? 5 km water + 6.5 km sediments? or do you mean that the bottom of the oceanic crust is 11.5 km beneath sea level? Or something else?*

**AC**: We agree to describe the structure more precisely. We defined the Moho at a depth of 11.5 km, meaning that the Moho is at 11.5 km below sea level (water depth is 4.9 km, oceanic crust thickness is 6.6 km). We adapted it accordingly in the main text.

*RC: L361: "Here only those events were used": Is this a subset of the 46 events you name above, or was this part of the selection criteria that led to 46 events being chosen. If the former, how many events were used finally?*

**AC**: This information is given in Table S1, which is referenced in line 370 in the main text. From all 46 events, some were used for SW analysis, some for RF analysis, and some for both. 9 out of the 46 events were used neither for the SW, nor the RF analysis.

*RC: L380: "improvements of the method" -> "improvements obtained using the method", I think.*

**AC**: It is now modified to "To quantify the improvements obtained when using our method".

*RC: L388: Does a high correlation coefficient really demonstrate that there is no waveform distortion? What is the threshold correlation coefficent for which this is true?*

**AC**: We agree that a high correlation coefficient solely doesn't demonstrate that there is no waveform distortion. Also there isn't any specific threshold for which the coefficient shows good preservation, since there is always some noise remaining after denoising and the amount of the remaining noise depends on the type of noise. However, a high correlation coefficient is an indication for signal preservation. Along with all the other tests, it helps to demonstrate the wanted earthquake waveform preservation. We adapted the text accordingly.

*RC: 394-396: Unclear.*
**AC**: We modified the unclear part and added more explanation to make it clear.

*RC: 403-4: Repeats what you already wrote.*
**AC**: We keep this since it is important to mention the peak on the arrival time of seismic phases and emphasis that the energy of seismic phases is preserved.

*RC: L406-7: Isn't 4D just an overlapping plot of the lines in Figure 4C? If so, you don't need to describe it in such detail.*

**AC**: Figure 4d is not overlapping plot of the lines in Figure 4C, but it is a comparison of the synthetic signal with the trace showing the "difference of SO and synthetic" and "difference of HPS and synthetic".

*RC: L438: The sentence starting with "Group velocity curves..." seems out of sequence.*

**AC**: This sentence is removed.

*RC: L440: "noise situations N1-N3": Be consistent in your naming, you refer to N1-N3 as "situations" here and in the figure captions, "scenarios" on line 357 and "type" on line 519.*

**AC**: Thanks for this remark. Now we used "situation" in all cases.

*RC: L445: "in the range of the signal frequencies" repeats, remove it. "0.05 to 0.2 Hz": you give a frequency range here but the figure only shows periods.*

**AC**: The dispersion maps show that noise energy in the range of the signal frequencies is removed successfully for periods between 5 and 20 s. Longer signal periods which are weakly visible in the noise-free image (Fig. 5d) can only partially be recovered.

*RC: L446: You state that longer signal periods can not be recovered, but it appears in the figure that they can for N1 and N3, as you state for N3 on lines 447-8.*

**AC**: We modified the sentence and mentioned that they can be only partially recovered.

*RC: L488-9: "became a broader peak..."Compared to SO? or to P_{410}S?*

**AC**: Compared to SO. We changed the sentence to clarify the comparison between SO and HPS.

*RC: L508-511: These sentences are not specific enough, they read more like a summary than a conclusion.*

**AC**: We modified these sentences and now they fit better in the conclusion section.

*RC: L514-515: Unless, I'm mistaken, this is the first time you mention extracting microseism signal. If so, this should be mentioned in the discussion, not the conclusion.*

**AC**: We mention this in the conclusion since this was not the purpose of the study, however this could be an application of this algorithm. We don't mention it in the discussion because we didn't specifically extract the microseism signal but one can do so by applying a band pass filter to the extracted noise signal.

*RC: L519-537: Much more specific and detailed than in the discussion, should be put in the discussion and simply referred to here.*

**AC**: We agree with the comment. We moved some details to the discussion and shortened the conclusion section.

*RC: L539-540: "and has especially application in noise reduction of OBS signals": seems just repeat the first half of the sentence.*

**AC**: The algorithm can extract and separate different signals in the OBS recordings. As mentioned in the previous comment, one can extract the microseism signal for further study on it. This is one application of this algorithm where the extracted signal is the wanted signal. Another application, which we focus on in this study, is noise reduction of OBS recordings where the extracted signals are considered as noise for the study of teleseismic earthquakes.

*RC: References and figures:*
Thanks for the suggestions and corrections. We applied all. We also added other missing DOIs.

---

## Editor Decision (ED1)

The authors generally reponded to my questions and I think the article is acceptable for publication.

There were a few cases where they stated that they responded to my request but the requested change is not in the manuscript:

*RC: L82: The IG wave signal used by Crawford and Webb was not recorded by a hydrophone, but by a differential pressure gauge. Differential pressure gauges, nano-precision bottom pressure recorders or broadband hydrophones can be used to measure the IG wave signal, though I'm not sure if broadband hydrphones are sensitive enough below their corner frequency.*
*AC: Thanks for this remark. We replaced "hydrophone data" by "differential pressure gauges".*
Not changed (L115 now).  Should replace "hydrophone data" by "pressure data".

*RC: L136-141: These details of the LOBSTER OBSs development aren't relevant to the method or the data presented.*
*AC: We removed this part from the manuscript.*
Not removed (L176-181 now)

*RC: L175-176: Repeats previous lines.*
*AC: The sentence is removed.*
Not removed (L216)

*RC: L 235: Why 2%? Is this a parameter you set? Or an observation of some separation in S-values?*
*AC: We use a threshold for picking the highest similarity. We choose the upper 2% of the time frames with highest S values as the similar frames. We modified the sentence and added the term "the upper" to make it clear.*
Actually it was already clear that this was the upper 2% (but the change in text is fine): my question was: why 2%?
Also, Make this a parameter in Table 1

*RC: Eq 7: Use the same emphasis in the equation as in the text (N and N' are bold in the text, but italicized in the equation)*
*AC: Within the whole manuscript, we used bold for the variables in the text and used italic for the equations.*
I don't see how this could be a good idea, but I leave it to the manuscript preparation team to decide.

*RC: L445: "in the range of the signal frequencies" repeats, remove it. "0.05 to 0.2 Hz": you give a frequency range here but the figure only shows periods.*
*AC: The dispersion maps show that noise energy in the range of the signal frequencies is removed successfully for periods between 5 and 20 s. Longer signal periods which are weakly visible in the noise-free image (Fig. 5d) can only partially be recovered.*
The response is not adapted to my comment, which was simply about 1) improving grammar and 2) avoiding inverse units between       sw s the text and the figure

The spectrogram plots still do not show axes units

**I also recommend the following changes:**

69-70: the sentence about projection of horizontal signals onto the vertical channel is not relevant to this article.

119: "subsequently" is not the right word: "in sequence" is one better option.

120 and 122: Remove "adopt HPS using"

129: "Loose cables" doesn't explain anything, especially if you are referring to the rope connecting the OBS to the recovery buoy, which is not technically "loose"

178: "In the context of HPS, one of the simplest and fastest approaches…" => One of the simplest and fastest HPS approaches…"

215: "In range two avoiding the frequency range of 0.1 to 1 Hz". This is redundant. Moreover, you refer to range one and range two here, but later on you explicitly name the frequency range and in Table 1 you refer to "Frequency range for MED" and "Frequency range for SIM". I recommend using "MED frequency range" and "SIM frequency range" everywhere, which will make the reading clearer and should also help the reader to understand the reason for separating these ranges.

255: Why do you use soft masks rather than a binary mask?

292: "waiting time" => "waiting factor" as on line 290.

304: "the mentioned frequency range" => "the MED frequency range" (or SIM, it's hard for me to tell/remember as it is currently written).

516: "MIR" => "Music Information Retrieval", as may readers will skip to this Conclusion

538-539: Remove this sentence, a "conclusion" of the "Conclusions" section is redundant

Table 1: Simplify parameter names and add the 2% term from line 238.

**And the following grammar corrections:**

*"is" used too often/inappropriately*
39: "is often dominating" => "often dominates"
51: "is originating" => "originating"
51: "is originating" => "originates"
138: "is highly affecting" => "highly affects"
288: "is separating" => "separates"

394: "it is an indication" => "they indicate"

*Overuse of "the":*
102: " music information"
253: " repeating and  nonrepeating"
297, 303 & 335: "the Sect. 2" =? "Section 2"
301: "the Sect. 3.2" =? "Section 3.2"
345: " Table 1"
384: "The  amplitude and  phase information…"

---

## Author Response (AR2)

**Response to the comments:**

*RC: L82: The IG wave signal used by Crawford and Webb was not recorded by a hydrophone, but by a differential pressure gauge. Differential pressure gauges, nano-precision bottom pressure recorders or broadband hydrophones can be used to measure the IG wave signal, though I'm not sure if broadband hydrphones are sensitive enough below their corner frequency.*
*Not changed (L115 now). Should replace "hydrophone data" by "pressure data".*

**AC**: Now it is replaced. (Sorry for the mismatch, it was an inadvertent mistake.)

*RC: L136-141: These details of the LOBSTER OBSs development aren't relevant to the method or the data presented.*
*Not removed (L176-181 now).*

**AC**: Now it is removed.

*RC: L175-176: Repeats previous lines.*
*Not removed (L216)*

**AC**: Now it is removed.

*RC: L 235: Why 2%? Is this a parameter you set? Or an observation of some separation in S-values?*
*AC: We use a threshold for picking the highest similarity. We choose the upper 2% of the time frames with highest S values as the similar frames. We modified the sentence and added the term "the upper" to make it clear.*
*Actually it was already clear that this was the upper 2% (but the change in text is fine): my question was: why 2%? Also, Make this a parameter in Table 1.*

**AC**: The parameter is added to the table. As it is mentioned in the text "We choose the upper 2%", so this is the parameter we set and it is a good suggestion to add it to the table.

*RC: Eq 7: Use the same emphasis in the equation as in the text (N and N' are bold in the text, but italicized in the equation)*
*AC: Within the whole manuscript, we used bold for the variables in the text and used italic for the equations.*
*I don't see how this could be a good idea, but I leave it to the manuscript preparation team to decide.*

**AC**: Now we use italic both in the text and equations.

*RC: L445: "in the range of the signal frequencies" repeats, remove it. "0.05 to 0.2 Hz": you give a frequency range here but the figure only shows periods.*

*AC: The dispersion maps show that noise energy in the range of the signal frequencies is removed successfully for periods between 5 and 20 s. Longer signal periods which are weakly visible in the noise-free image (Fig. 5d) can only partially be recovered.*
*The response is not adapted to my comment, which was simply about 1) improving grammar and 2) avoiding inverse units between the text and the figure.*

**AC**: We modified the text and improved it. Also we modified the figure so now we use Hz as the unit of frequency both in the figures and in the text.

*RC: Figure 2b: put units on axes of spectrogram plots*
*AC: Thanks for the suggestions and corrections. We applied all….*
*The spectrogram plots still do not show axes units.*

**AC**: The plot is modified and all the units are now shown on the axes.

*RC: 69-70: the sentence about projection of horizontal signals onto the vertical channel is not relevant to this article.*
**AC**: The sentence is now removed.

*RC: 119: "subsequently" is not the right word: "in sequence" is one better option.*

**AC**: Now we replaced it.

*RC: 120 and 122: Remove "adopt HPS using"*
**AC**: We removed it.

*129: "Loose cables" doesn't explain anything, especially if you are referring to the rope connecting the OBS to the recovery buoy, which is not technically "loose".*
**AC**: We modified the sentence and remove the term "loose".

*RC: 178: "In the context of HPS, one of the simplest and fastest approaches…" => One of the simplest and fastest HPS approaches…"*
**AC:** The sentence is modified according to the comment.

*RC: 215: "In range two avoiding the frequency range of 0.1 to 1 Hz". This is redundant. Moreover, you refer to range one and range two here, but later on you explicitly name the frequency range and in Table 1 you refer to "Frequency range for MED" and "Frequency range for SIM". I recommend using "MED frequency range" and "SIM frequency range" everywhere, which will make the reading clearer and should also help the reader to understand the reason for separating these ranges.*
AC: That is a good suggestion. We use the suggested term now.

*RC: 255: Why do you use soft masks rather than a binary mask?*
**AC:** Soft masks are more flexible than binary masks and usually lead to better results; it's not very often that all of the energy in a mixture can always be assigned to one

source. We added a brief explanation about this to the text and more detail can be found in the related reference (Vaseghi, 1996).

*RC: 292: "waiting time" => "waiting factor" as on line 290.*
**AC**: Now we replaced it.

*RC: 304: "the mentioned frequency range" => "the MED frequency range" (or SIM, it's hard for me to tell/remember as it is currently written).*
**AC**: We use the suggested term now.

*RC: 516: "MIR" => "Music Information Retrieval", as may readers will skip to this Conclusion*
**AC**: We added it.

*RC: 538-539: Remove this sentence, a "conclusion" of the "Conclusions" section is redundant*
**AC**: The sentence is removed.

*RC: Table 1: Simplify parameter names and add the 2% term from line 238.*
**AC**: We simplified the parameters and added the 2% term.

All the grammar correction are applied according to the suggestions.